# Assessment of Urethral Elasticity by Shear Wave Elastography: A Novel Parameter Bridging a Gap Between Hypermobility and ISD in Female Stress Urinary Incontinence

**DOI:** 10.3390/bioengineering12040373

**Published:** 2025-04-01

**Authors:** Desirèe De Vicari, Marta Barba, Clarissa Costa, Alice Cola, Matteo Frigerio

**Affiliations:** Department of Gynecology, IRCCS San Gerardo dei Tintori, University of Milano-Bicocca, 20900 Monza, Italy; d.devicari@campus.unimib.it (D.D.V.); m.barba8792@gmail.com (M.B.); c.costa14@campus.unimib.it (C.C.); alice.cola1@gmail.com (A.C.)

**Keywords:** urethral elasticity, bladder neck descent (BND), shear wave elastography (SWE)

## Abstract

Stress urinary incontinence (SUI) results from complex anatomical and functional interactions, including urethral mobility, muscle activity, and pelvic floor support. Despite advancements in imaging and electrophysiology, a comprehensive model remains elusive. This study employed shear wave elastography (SWE), incorporating sound touch elastography (STE) and sound touch quantification (STQ) with acoustic radiation force impulse (ARFI) technology, to assess urethral elasticity and bladder neck descent (BND) in women with SUI and continent controls. Between October 2024 and January 2025, 30 women (15 with SUI, 15 controls) underwent transperineal and intravaginal ultrasonography at IRCCS San Gerardo. Statistical analysis, conducted using JMP 17, revealed significantly greater BND in the SUI group (21.8 ± 7.8 mm vs. 10.5 ± 5 mm) and increased urethral stiffness (Young’s modulus: middle urethra, 57.8 ± 15.6 kPa vs. 30.7 ± 6.4 kPa; *p* < 0.0001). Mean urethral pressure was the strongest predictor of SUI (*p* < 0.0001). Findings emphasize the role of urethral support and connective tissue integrity in continence. By demonstrating SWE’s diagnostic utility, this study provides a foundation for personalized, evidence-based approaches to SUI assessment and management.

## 1. Introduction

Stress urinary incontinence (SUI) is defined as the involuntary leakage of urine during activities that increase intra-abdominal pressure, such as coughing, sneezing, laughing, or engaging in physical exertion. It is one of the most prevalent forms of urinary incontinence among women and represents a significant public health concern due to its impact on physical, psychological, social, and economic well-being. The condition arises from a dysfunction of the pelvic floor muscles and the urethral sphincter, resulting in impaired support and compromised urethral closure during episodes of increased abdominal pressure [1].

It is estimated to affect approximately 46% of adult women. This statistic highlights its widespread prevalence, making it a common issue that significantly impacts the quality of life and daily functioning for nearly half of the female adult population [2].

The etiology is multifactorial, with vaginal delivery being a key risk factor due to pelvic floor trauma. Additional contributors include aging, menopause, high BMI, and prior pelvic surgeries like hysterectomy. Post-menopausal estrogen loss and lifestyle factors such as chronic coughing, smoking, and heavy lifting further weaken pelvic floor muscles, increasing SUI risk [2].

Stress urinary incontinence has a profound impact on women’s quality of life, affecting both physical functioning and psychological well-being. Women with SUI often experience embarrassment, anxiety, and a loss of self-confidence due to the unpredictable nature of urine leakage. This can lead to avoidance of social interactions, decreased participation in physical activities, and a diminished overall sense of well-being. In more severe cases, SUI can contribute to social isolation, depression, and a deterioration in mental health [3].

The economic implications of SUI are also considerable. Women affected by the condition frequently rely on absorbent pads, protective garments, and other hygiene products to manage their symptoms, which can result in substantial ongoing expenses. Moreover, the healthcare costs associated with the diagnosis and treatment of SUI—including physiotherapy, pharmacologic interventions, and surgical procedures—add to the financial burden faced by both individuals and healthcare systems [4].

Despite its prevalence and significant impact, SUI remains underdiagnosed and undertreated. Many women do not seek medical care due to embarrassment, a lack of awareness about available treatment options, or the misconception that SUI is a natural consequence of aging or childbirth. However, evidence-based interventions, including pelvic floor muscle training, lifestyle modifications, and surgical options, can significantly improve symptoms and quality of life for affected individuals [5].

Given its physical, psychological, and economic consequences, stress urinary incontinence in women warrants increased attention from healthcare providers, policymakers, and researchers. A comprehensive approach to prevention, early identification, and management is essential to mitigate the impact of SUI and enhance the overall health and well-being of women across the lifespan.

### 1.1. Anatomy and Physiology of the Female Urogenital Diaphragm

The mechanisms responsible for maintaining urethral closure at rest and during periods of increased abdominal pressure are well established, though their interrelationships remain complex and not fully understood. Key factors contributing to urethral closure include a healthy striated sphincter innervated by the pudendal nerve, well-vascularized urethral mucosa and submucosa, properly functioning intrinsic urethral smooth muscle, and intact urethro-vaginal support structures [1].

Historically, numerous theories have been proposed to explain stress urinary incontinence (SUI). Recent scientific studies comparing women with and without SUI suggest that both weakened urethral sphincter function and impaired bladder and urethral support mechanisms play significant roles in the condition [6]. Damage to the pelvic floor muscles, bladder neck, and urethral sphincters further contributes to the pathophysiology of SUI.

The urethra is anatomically supported by the anterior vaginal wall, with key points of attachment at the superior vaginal sulcus, where the vaginal “hammock” connects to the pubococcygeus muscle. These structures, along with the arcus tendineus fascia pelvis—also referred to as pubo-urethral ligaments—anchor the urethra to the pubic bone. These fascial structures maintain strength over time, but elongation or damage to them has been hypothesized to contribute to urethral support loss in SUI, though objective evidence remains limited [7].

The striated urogenital sphincter (SUS), also referred to as the external urethral sphincter (EUS), plays a critical role in maintaining urethral closure. Anatomical studies have identified the SUS as a superior horseshoe-shaped structure surrounding the urethra, with an inferior component encircling the anterolateral urethra and lateral vagina. The levator ani muscle (LAM) also contributes to urethral support, particularly through its attachment to the SUS. This tendinous connection facilitates anterior bending of the mid-urethra during simultaneous contraction of the LAM and SUS, promoting urethral closure and continence [8].

Pelvic floor ultrasound studies have further demonstrated the role of anatomical variations in urethral support. In a retrospective study of women with urodynamic stress incontinence, a longer genitohiatal distance was associated with reduced functional urethral parameters, such as lower maximum urethral closure pressure and Valsalva leak-point pressure. However, these anatomical differences account for only a small portion of the variability observed in SUI [9].

In summary, the maintenance of urethral closure involves a complex interplay of anatomical and functional factors. Damage to the pelvic floor muscles, ligaments, and sphincter mechanisms, combined with impaired motor control, contributes to the development of SUI. While various structures, including the urethral sphincter, vaginal wall, and levator ani muscles, play a role in continence, the precise contributions and interactions of these elements remain an area of ongoing research.

### 1.2. Pathophysiology of Stress Urinary Incontinence in Women

The pathophysiology of stress urinary incontinence presents a complex interplay of anatomical, physiological, and neurological factors that must be addressed in both clinical practice and research. Continuous evaluation and integration of emerging evidence will enhance the understanding, prevention, and treatment of SUI, ultimately improving quality of life for affected women. A weakened urethral sphincter reduces closure pressure, compromising its ability to prevent involuntary urine leakage. Additionally, the weakening of supportive structures and the loss of normal pelvic support contribute to the progression of incontinence. Comparative studies between continent and incontinent women underscore the importance of assessing both anatomical integrity and functional capacity in understanding the mechanisms underlying SUI [10].

The etiology of SUI is multifactorial, with vaginal delivery being a primary risk factor due to the potential trauma to the pelvic floor muscles, connective tissues, and nerves that can occur during childbirth. Other contributing factors include advancing age, menopause, elevated body mass index (BMI), and a history of pelvic surgery, particularly hysterectomy. Age-related changes, such as the loss of estrogen following menopause, further contribute to a decline in pelvic floor integrity, exacerbating the risk of SUI [4]. Additionally, lifestyle factors such as chronic coughing, smoking, and repetitive heavy lifting can place additional strain on the pelvic floor muscles, increasing the likelihood of developing the condition [1].

The primary causes of stress urinary incontinence (SUI) in the female population are explored in detail below.

#### 1.2.1. Childbirth and Its Effects

Vaginal childbirth can impact pelvic anatomy and nerve function, increasing the risk of stress urinary incontinence (SUI). Damage to the pudendal nerve, levator ani muscles, and vaginal walls weakens pelvic support, raising urethral mobility and reducing closure pressure [11]. Studies show vaginal delivery carries a 67% higher long-term SUI risk than cesarean, with forceps delivery further increasing this risk [12,13].

While cesarean offers some protection, this benefit decreases with time [12]. Postpartum SUI often resolves within a year but can predict future incontinence due to lasting pelvic floor trauma. Risk factors include advanced maternal age, high BMI, diabetes, and forceps delivery [14,15,16]. The protective effect of cesarean remains uncertain, particularly regarding elective procedures before labor. [17].

Pelvic floor injuries during vaginal birth result from nerve, muscle, and connective tissue damage, particularly affecting the pudendal nerve [18]. Levator ani muscle trauma is common but inconsistently linked to SUI [10]. Nerve injuries reduce urethral closure pressure and persist even after recovery [19,20]. Epidurals and episiotomies show no significant effect on postpartum incontinence [21,22].

#### 1.2.2. Role of Pelvic Organ Prolapses

Pelvic organ prolapse (POP) contributes to stress urinary incontinence (SUI) by causing bladder outlet obstruction, detrusor overactivity, and latent SUI, which can emerge post-POP surgery. Studies report that 40–50% of women undergoing POP repair develop SUI, with varying degrees of severity [23,24]. While urodynamic studies (UDSs) can identify latent SUI preoperatively, their impact on surgical planning is debated [25]. Managing SUI alongside POP remains controversial, though evidence suggests that adding a mid-urethral sling (MUS) during POP repair reduces postoperative SUI. However, the optimal timing remains uncertain [26].

#### 1.2.3. Urethral Position and Support

SUI is associated with anatomical and functional urethral changes. MRI studies confirm a thinner urethral sphincter in SUI patients [27,28]. Functional urethral length is consistently shorter in women with SUI, correlating with symptom severity [10]. Bladder neck funneling is significantly more frequent in SUI patients, occurring up to five times more than in continent women [10]. Periurethral ligament defects, found in 76% of SUI cases, contribute to incontinence risk [27,29,30]. Increased urethral mobility, reflected in larger α and β angles, is also linked to SUI [10,31,32,33].

#### 1.2.4. The Role of Obesity

Obesity is a major SUI risk factor, primarily due to increased intra-abdominal pressure, which raises intravesical pressure and stresses the pelvic floor muscles [34,35]. Urodynamic studies confirm that obese women have higher intravesical pressures during rest and coughing [36,37]. Additionally, obesity-related neuropathy may impair pelvic floor nerve function [38,39,40]. Animal studies suggest obesity weakens urethral sphincter contractility, further exacerbating SUI risk [41,42].

#### 1.2.5. Alteration of Connective Tissue and Urethral Elasticity

Collagen abnormalities play a critical role in SUI pathophysiology. SUI patients exhibit reduced total collagen and altered matrix metalloproteinase activity, leading to weakened connective tissue [43,44,45]. Aging, pregnancy, and hormonal changes contribute to these structural defects [46,47]. While elastin degradation has been less studied, abnormal elastin turnover in women with prolapse may impact urethral support [48,49].

#### 1.2.6. Urethral Dysfunction and Intrinsic Sphincter Deficiency (ISD)

The 2017 Seventh International Consultation on Incontinence and NIH symposia highlighted urethral insufficiency as a key contributor to stress incontinence [50]. Blaivas et al. initially classified stress incontinence into three types based on urethral mobility, with Type III later evolving into the modern concept of ISD. This term focuses on intrinsic urethral factors such as pudendal innervation, sphincter mass, and mucosal integrity, independent of vaginal mobility [51]. ISD was initially used to explain surgical failures and incontinence in the absence of vaginal movement. It is characterized by low urethral closure pressure, a “stovepipe” urethral appearance, and funneling under minimal pressure. Common causes include surgical injury, ischemia, and radiation damage [1,50].

#### 1.2.7. Hypermobility and Intrinsic Sphincter Deficiency: Moving Beyond a Binary Framework to a Continuum Approach

There is now a shift away from the hypermobility–ISD binary model toward a continuum perspective. Valsalva leak point pressure (VLPP) has emerged as a tool for assessing urethral function, showing that reduced urethral closure pressure accounts for ~50% of stress incontinence, with urethral support adding 11% to predictive accuracy [52]. VLPP studies on urethral bulking revealed that continence post-treatment is linked more to leakage pressure than urethral closure pressure [52]. Furthermore, surgical outcome studies suggest that hypermobility patients may have underlying ISD, explaining higher failure rates in certain procedures [52].

Research by Horbach and Ostergaard found age to be an independent predictor of ISD, linking muscle loss and slowed reflexes to the condition [53]. Perucchini et al. demonstrated a decline in urethral striated muscle fibers with aging, correlating with a 15% annual drop in urethral closure pressure in nulliparous women [54,55]. Kayigil et al. noted that ISD prevalence in hypermobility patients likely contributes to the high failure rates of bladder neck suspension procedures [56]. Consequently, suburethral sling surgery is increasingly recommended as the primary treatment for stress incontinence, rather than being reserved for recurrent cases [57].

### 1.3. Literature Review

Stress incontinence results from complex interactions among anatomical, functional, and genetic factors, including urethral position, muscle activity, and pelvic floor support. Despite advancements in imaging and electrophysiology, a comprehensive model remains elusive. Further research is needed to clarify the relationships between hypermobility, intrinsic sphincter deficiency (ISD), and pudendal nerve function.

### 1.4. Studies of Urethral Position

Urethral mobility is a key factor in stress urinary incontinence (SUI), as confirmed by studies from Hodgkinson, Jeffcoate, and Roberts [58,59]. Animal models support this, showing that trauma to urethral supports like the pubourethral ligaments leads to SUI [60]. Surgical stabilization of urethral position improves continence, though the role of active muscle contraction remains unclear [61].

### 1.5. Studies of Urethral Pressure and Resistance

SUI is associated with reduced urethral pressure and closure resistance, though correlations between maximum urethral closure pressure (MUCP) and Valsalva leak point pressure (VLPP) remain debated. Almeida et al. found a significant relationship, while Martan et al. did not [62,63]. Urethral resistance pressure (URP) is also reduced in SUI, with ultrasound studies linking low URP to decreased muscle layers [64,65].

### 1.6. Electrophysiological Studies of Urethral Function

Nerve dysfunction plays a role in SUI, as shown by Snooks and Swash [66,67]. Prolonged pudendal nerve terminal motor latency and urethral muscle dysfunction contribute to incontinence [68]. Animal models confirm that pudendal nerve injury lowers LPP, leading to leakage [69]. Electromyographic studies highlight altered muscle responses in SUI patients, including decreased MUCP after repeated coughing [70,71]. Post-surgical incontinence is linked to poorer neuromuscular function [72].

### 1.7. Genetic Factors

Chen et al. identified elastin metabolism gene variations in women with SUI, suggesting a role for connective tissue remodeling [73]. MicroRNA-214 also influences fibroblast differentiation in pelvic floor dysfunction models [74]. However, the impact of genetic factors on urethral tissue remains uncertain

### 1.8. Role of Advanced Imaging in Understanding Pathophysiology

#### 1.8.1. Magnetic Resonance Imaging

MRI provides dynamic visualization of pelvic structures and superior soft tissue resolution, aiding in the assessment of urethral mobility and prolapse [74]. Functional MRI has shown cortical activation during pelvic floor muscle training [75,76].

#### 1.8.2. Real-Time Ultrasonography

Ultrasound offers cost-effective, real-time visualization of urethral function. Studies show urethral funneling is linked to lower closure pressures and increased ISD risk [77]. 3D ultrasound has revealed differences in urethral movement under stress [78].

### 1.9. Aim of the Study

Stress incontinence results from a multifaceted interplay of anatomical and functional factors, including urethral positioning, muscular activity, and the structural support of the pelvic floor. Extensive research has been undertaken to elucidate key components such as urethral closure mechanisms, pressure dynamics, and electromyographic (EMG) activity, with the objective of advancing the understanding of conditions such as stress urinary incontinence (SUI) and pelvic organ prolapse in women. Despite notable advancements in electrophysiological methodologies and imaging technologies, a unified and comprehensive model integrating these interdependent elements has yet to be developed.

Investigations into urethral function underscore the pivotal roles of urethral mobility and pelvic floor muscle performance. Advanced imaging modalities, including dynamic magnetic resonance imaging (MRI) and ultrasonography, have yielded valuable insights into urethral behavior under stress. However, the precise interrelationships among urethral hypermobility, intrinsic sphincter deficiency, and neural function remain insufficiently defined. Addressing these knowledge gaps is essential for the development of targeted and efficacious therapeutic interventions for stress incontinence.

The present study sought to examine differences in urethral elasticity and bladder neck descent (BND) between women with stress urinary incontinence (SUI) and continent controls, utilizing shear wave elastography (SWE) as an innovative diagnostic technique. SWE facilitated quantitative assessments of these parameters, providing critical insights into the structural and functional alterations associated with SUI. By focusing on urethral elasticity and BND, this study aimed to enhance the understanding of the pathophysiological mechanisms underlying SUI and to evaluate the potential clinical utility of SWE as a diagnostic tool to improve management strategies for this condition.

## 2. Materials and Methods

### 2.1. Patients

This prospective observational comparative study involved two groups of participants: a group of women with stress urinary incontinence (SUI) and a control group of healthy continent women. The participants underwent pelvic ultrasound examinations at the gynecological ultrasound outpatient clinic of the Gynecological Department, IRCCS San Gerardo, between October 2024 and January 2025. The clinical evaluation was conducted by our urogynecology team, which possesses decades of experience in managing this condition. The ultrasound evaluation was carried out by a team member who holds a master’s degree in pelvic floor ultrasound.

Inclusion criteria: For the group of women with stress urinary incontinence (SUI), the inclusion criteria were based on a history of involuntary urine leakage during increased intra-abdominal pressure, supported by a positive stress test with a full bladder (at least 300 mL). For the control group, participants were women who had never experienced episodes of stress urinary incontinence and reported no symptoms of urinary incontinence.

Exclusion criteria included prior gynecological or urological surgeries (e.g., removal of ovaries, uterus, or vagina; urethral surgeries; placement of urethral prostheses), pregnancy or delivery within the past 12 months, pelvic organ prolapse, age under 18 years, active infections, post-void residual urine exceeding 150 cc, voiding difficulties, or use of medications affecting continence. Women with stress urinary incontinence were included based on subjective symptoms of involuntary urine leakage during increased intra-abdominal pressure, supported by a positive stress test with a full bladder. All participants underwent gynecological evaluations to exclude pelvic organ prolapse.

### 2.2. Ultrasound Examination

Ultrasonography was performed using the Mindray Resona I9 with ZONE Sonography^®^ Technology+ (ZST+) in the lithotomy position. Patients were instructed to empty their bladder before the procedure. The ultrasound assessments employed both transperineal and intracavitary–intravaginal approaches.

For transperineal ultrasonography, measurements were taken by positioning the convex probe (SC 6-1s, Mindray Resona I9; frequency 1.2–5.2 MHz, field of view: 55°) on the symphysis pubis to capture sagittal views, visualizing the symphysis pubis, bladder, and urethra. Key anatomical axes, including the central axis of the symphysis pubis, proximal urethral axis, posterior bladder wall axis, and a horizontal line crossing the posteroinferior margin of the symphysis pubis, were marked. Bladder neck descent (BND) was determined by measuring the distance between the bladder neck and the horizontal axis passing through the distal symphysis pubis during rest and Valsalva (Figure 1).

For intravaginal evaluation, a biplanar high-frequency intracavity probe (ELC13-4 s, Mindray Resona I9; linear array probe, frequency 3.50–9.5 MHz, scanning depth 1.5–35 cm) equipped with SWE software was used.

The probe, covered with a protective sleeve and gel, was gently inserted into the vagina. The linear array probe was oriented with the marker directed to 12 o’clock, ensuring the sound beam was perpendicular to the urethral wall. Images were deemed satisfactory when the bladder and urethral air lines were distinctly visualized (Figure 2). Measurements were then documented.

STE real-time shear wave elastography (SWE) was employed during the examination, with patients instructed to hold their breath to minimize movement. The examiner maintained a steady hand while activating the triple-image real-time display function, allowing simultaneous observation of the 2D ultrasound image, its corresponding elastography image, and the reliability of the elastography RLB Map. Reliability indicators for the optimal shear wave elastography (SWE) frame included the presence of a fully green RLB Map and a five-green-star Motion Stability Index (Figure 3).

Regions of interest (ROIs) with a 2 mm diameter were identified at the proximal, middle, and distal segments of the urethral wall. Measurements were conducted in six defined regions: the anterior wall at sections positioned approximately 1 cm proximal to the bladder neck, 1 cm from the mid-urethral segment, and 1 cm distal near the external urethral meatus, along with the corresponding posterior wall sections at these specified locations (Figure 4).

The Young’s modulus (E) values, representing tissue stiffness, were recorded for each ROI. In soft tissues, Young’s modulus is significantly lower compared to engineering materials (E ≈ 1 MPa). In both humans and mammals, mechanical behavior is largely governed by a limited set of extracellular matrix (ECM) proteins, predominantly fibrillar proteins such as type I collagen, fibrillin, and elastin. While elastin is more compliant, fibrillar proteins in their structural form are notably stiffer than the surrounding tissue. Generally, the elastic stiffness of tissue components tends to decrease as the structural scale increases.

From the collected data, the mean Young’s modulus (Emean) of the urethral wall was calculated, providing a quantitative measure of its hardness and serving as an index for evaluating urethral wall stiffness.

### 2.3. Statistical Analysis

Statistical analyses were performed using JMP 17 (SAS Institute, Cary, NC, USA). Descriptive statistics were calculated to summarize the data: categorical variables were presented as absolute frequencies and percentages, while continuous variables were expressed as means ± standard deviations.

To evaluate differences between groups, paired *t*-tests were applied for continuous variables, ensuring that comparisons accounted for paired data when appropriate. The statistical significance threshold was set at *p* < 0.05, with values below this level indicating significant differences. A generalized linear model using logistic regression was developed to analyze the relationship between three predictive variables of urethral pressure (mean, distal, and proximal) and the likelihood of belonging to the diseased or healthy group, highlighting the statistical significance of mean and distal urethral pressure. All computations adhered to standard statistical practices to ensure the robustness and reliability of the results.

## 3. Results

### 3.1. Demography

Within the designated timeframe, a total of 30 patients meeting the inclusion criteria underwent ultrasound evaluation. The cohort was divided into two groups: Group 1 comprised 15 patients diagnosed with stress urinary incontinence (SUI), while Group 2, serving as the control group, included 15 continent individuals without symptoms or objective evidence of SUI. The demographic and clinical characteristics of these two populations were analyzed.

In Group 1, the patients had a mean age of 56 years. The average parity was 1.6, with 13% of patients being nulliparous, 26% having one child, 46.6% having two children, and 6.6% having either three or four children. Slightly more than half of the patients in this group were menopausal and not undergoing hormone replacement therapy. Group 2 had a mean age of 49.5 years. Menopausal status and parity were found to overlap within Group 1 (SUI). For both groups, the mean body mass index (BMI) fell within the normal range, with Group 1 reporting a mean BMI of 23.7 and Group 2 a mean BMI of 19.9.

### 3.2. Comparison of BND and Young’s Modulus Among the Groups

A comparative analysis between the two groups was performed using an independent-sample *t*-test, demonstrating a statistically significant difference in bladder neck descent under maximal Valsalva. The group with stress urinary incontinence (SUI, Group 1) exhibited a greater descent compared to the continent population (Group 1: 21.8 ± 7.8 mm, Group 2: 10.5 ± 5 mm; *p* < 0.0001) (Table 1).

In addition, the evaluation of Young’s modulus values revealed that the continent population had significantly higher-pressure values (in kPa) in both the middle and distal portions of the urethra compared to the SUI group (Table 2). These differences were consistent when analyzing the anterior and posterior walls individually and collectively. In the middle portion of the urethra, the Young’s modulus values for the continent group were 30.7 ± 6.4 kPa for the anterior wall and 30.9 ± 5.5 kPa for the posterior wall, compared to 57.8 ± 15.6 kPa and 55.9 ± 15.6 kPa, respectively, in the SUI group (*p* < 0.0001 for all comparisons). Similarly, in the distal portion, the continent group exhibited Young’s modulus values of 29.9 ± 6.8 kPa for the anterior wall and 29.9 ± 7.1 kPa for the posterior wall, in contrast to 47 ± 7.8 kPa and 48 ± 10.9 kPa in the SUI group (*p* < 0.0001 for all comparisons). When the anterior and posterior walls were analyzed together, the continent group showed average values of 30.8 ± 5.8 kPa in the middle portion and 29.9 ± 6.8 kPa in the distal portion, compared to 56.8 ± 15.4 kPa and 47.5 ± 9.4 kPa in the SUI group (*p* < 0.0001 for both comparisons) (Table 1 and Table 2).

These results indicate a significant disparity in bladder neck descent and urethral elasticity between the two groups, with the SUI population showing greater mobility and tendency for urethral deformation under stress.

In the generalized linear model designed to examine the binary variable distinguishing between the diseased and healthy populations, three predictors related to Young’s modulus of urethral pressure were analyzed: M (mean urethral pressure), D (distal urethral pressure), and P (proximal urethral pressure). The findings reveal that the overall model is statistically significant, with a Chi-square L-R value of 40.7066 and a *p*-value of <0.0001, indicating that at least one of the predictor variables significantly contributes to differentiating between the two groups.

Upon analyzing the individual effects, mean urethral pressure (M) emerges as the most influential predictor, with a highly significant *p*-value of <0.0001. The estimated coefficient for M is negative (estimate = −1.895073), indicating that higher mean urethral pressure is associated with a lower likelihood of belonging to the diseased group. Distal urethral pressure (D) also proves significant, with a *p*-value of 0.0087 and a negative coefficient (estimate = −0.519085), suggesting that an increase in this variable reduces the risk of stress urinary incontinence (SUI). In contrast, proximal urethral pressure (P) is not statistically significant, with a *p*-value of 0.4358, indicating that it does not meaningfully contribute to distinguishing between the two populations.

While the model demonstrates a strong fit to the data, the goodness-of-fit statistics reveal certain limitations. Pearson’s deviance value is 0.8822, with a *p*-value of 1.0000, suggesting an almost complete separation between the healthy and diseased data points. This condition, further confirmed by the Hessian matrix, highlights the potential for unstable parameter estimates and limits the model’s generalizability. Although the model fits the observed data well, it may suffer from overfitting, which could impair its predictive performance when applied to new datasets.

In summary, the results identify mean urethral pressure (M) as the most significant factor in distinguishing between healthy and diseased groups, followed by distal urethral pressure (D), while proximal pressure (P) does not appear to play a meaningful role.

## 4. Discussion

This study investigated differences in urethral elasticity and bladder neck descent (BND) between women with stress urinary incontinence (SUI) and continent controls using shear wave elastography (SWE). The findings revealed significantly greater BND and reduced urethral elasticity in the SUI group, reinforcing the existing literature on the role of altered urethral support and elasticity in SUI pathophysiology.

### 4.1. Bladder Neck Descent (BND)

This study demonstrated a statistically significant increase in BND during maximal Valsalva maneuver among women with SUI compared to controls (21.8 ± 7.8 mm vs. 10.5 ± 5 mm, *p* < 0.0001). This aligns with previous research showing that increased BND is linked to weakened pelvic floor support and contributes to urethral hypermobility, a primary factor in SUI development. The association between pelvic floor trauma and increased BND suggests that childbirth-related damage to the pubourethral ligaments and levator ani muscles may exacerbate urethral instability [1,15,16,18].

The observed increase in BND among SUI patients highlights the necessity of maintaining adequate urethral support to ensure continence. Surgical interventions such as mid-urethral slings, which aim to limit bladder neck mobility, have been shown to effectively restore continence, further supporting the clinical significance of these findings.

BND assessments were performed with an empty bladder, as previous research by Dietz et al. demonstrated that bladder filling reduces mobility, potentially leading to lower BND values [79]. While studies suggest that imaging the bladder neck with an empty bladder may be more challenging, the approach recommended by Pizzoferrato et al. and Alper Turkoglu et al. was followed to ensure optimal conditions for evaluating BND [80,81].

Several studies have investigated the role of ultrasonography in assessing BND and its correlation with SUI. Xiao et al. used transperineal 3D ultrasonography to evaluate SUI, reporting a mean BND of 2.19 ± 0.80 cm in the SUI group and 1.14 ± 0.66 cm in the control group (*p* < 0.001). They identified a BND threshold of 24 mm, with a sensitivity of 66.4% and specificity of 84.5%. Their findings suggest that while transperineal ultrasonography alone may not be sufficient to predict SUI, it can reduce unnecessary urodynamic testing by identifying cases unlikely to involve SUI [82].

Similarly, Naranjo-Ortiz et al. found that a BND greater than 25 mm indicated urethral hypermobility, with a sensitivity of 58% and specificity of 60% [83]. Hajebrahimi et al. reported a mean BND of 15.64 ± 9.65 mm in the SUI group and 8.13 ± 9.16 mm in controls, with a statistically significant difference (*p* < 0.01) [84]. Li et al. also evaluated transperineal ultrasonography for SUI diagnosis, reporting BND values consistent with those identified by Xiao et al. (*p* < 0.001) [85].

In the study by Alper Turkoglu et al., the mean BND in the SUI group was 16.6 ± 4.22 mm, compared to 6.53 ± 1.69 mm in the control group. The threshold BND was established at 11.2 mm, with assessments performed at a bladder volume of 150–300 cc [81]. These findings support the reliability, non-invasiveness, and practicality of ultrasonography as a dynamic imaging modality for assessing urethral mobility.

Our study reported a mean BND of 21.86 mm in the SUI group, consistent with values found in prior research. Notably, our findings deviated more from the average BND values in studies where the bladder was not completely empty during evaluation. This underscores the importance of standardized bladder volume conditions when assessing BND to ensure accurate comparisons across studies.

### 4.2. Urethral Elasticity and Young’s Modulus

Significant tendency for urethral deformation under stress, quantified by lower Young’s modulus values, was another notable finding. The SUI group exhibited significantly lower elasticity in both the middle and distal portions of the urethra when compared to controls (middle: 30.8 ± 5.8 kPa vs. 56.8 ± 15.4 kPa, *p* < 0.0001; distal: 29.9 ± 6.8 kPa vs. 47.5 ± 9.4 kPa, *p* < 0.0001). These results align with prior studies suggesting that changes in connective tissue composition and structural integrity play a pivotal role in the pathogenesis of SUI.

Reduced elasticity may be attributed to alterations in extracellular matrix (ECM) proteins such as collagen and elastin. Women with SUI often exhibit decreased collagen content, abnormal collagen cross-linking, and increased matrix metalloproteinase activity, which degrade ECM components and weaken urethral and pelvic floor tissues. Hormonal changes, particularly post-menopause, exacerbate these effects by further reducing collagen synthesis and elasticity [45,46,47,48,49].

Ultrasonic shear wave elastography (SWE) is a non-invasive imaging modality designed to quantitatively assess tissue stiffness through the use of shear waves [86]. This technology has been successfully utilized for evaluating the stiffness of various tissues, including the liver, kidney, thyroid, and muscles. Two advanced SWE methods, sound touch elastography (STE) and sound touch quantification (STQ), employ acoustic radiation force impulse (ARFI) technology to generate shear waves, track their propagation, and measure tissue displacement within a defined region of interest (ROI).

STQ is a point-specific SWE technique that uses ARFI to assess a small ROI, providing a quantitative measurement of stiffness, where higher shear wave velocity (SWV) reflects increased tissue stiffness. It typically measures the average rather than the maximum elastic modulus of the ROI [87]. In contrast, STE is a bi-dimensional SWE (2D-SWE) method that applies ARFI over an extended ROI to generate a real-time color-coded stiffness map, enabling calculation of SWV and the maximum elastic modulus. These advanced technologies enhance the precision, speed, and comprehensiveness of shear wave recordings compared to traditional SWE techniques [88].

This diagnostic technique has recently been applied to the field of gynecology and pelvic floor assessment, offering valuable insights into the mechanical properties of pelvic floor structures. Specifically, it employs Young’s modulus to quantitatively evaluate the elasticity of various anatomical components, including the vagina and the perineal body [89,90]. By measuring tissue stiffness, this method facilitates the analysis of elasticity variations across diverse populations, contributing to a deeper understanding of structural differences and potential clinical implications in pelvic floor health.

Continence, particularly in the urethra, is primarily influenced by the interplay of muscular, neural, and anatomical functions, including sphincter muscle tone and contraction, urethral tissue compliance, and the pressure exerted by surrounding structures. However, Young’s modulus may indirectly contribute to continence by determining the tissue’s ability to stretch or compress without compromising functional integrity. For instance, in the urethra, an optimal Young’s modulus supports maintaining the lumen closure or regulating its opening under pressure. The significant differences in Young’s modulus observed between the SUI and control groups reinforce the hypothesis that urethral stiffness is a critical determinant of continence. By offering quantitative insights, this study further supports the adoption of shear wave elastography (SWE) as a diagnostic tool for evaluating urethral function and informing treatment strategies.

### 4.3. Implications for Clinical Practice

The findings from this study have several significant implications for clinical practice. Shear wave elastography (SWE) presents a non-invasive, reproducible method for quantifying urethral stiffness and assessing bladder neck mobility. By incorporating SWE into clinical settings, diagnostic accuracy can be enhanced, which would allow for more personalized treatment strategies for stress urinary incontinence (SUI). Furthermore, the study suggests that reduced urethral elasticity could serve as a predictor of surgical outcomes. Patients with markedly reduced elasticity may benefit more from interventions targeting intrinsic sphincter deficiency, such as urethral bulking agents, rather than procedures focused on urethral support. Additionally, understanding the role of extracellular matrix (ECM) alterations in SUI highlights potential therapeutic targets. Future treatments may focus on modulating the turnover of collagen and elastin, possibly through pharmacological agents or regenerative medicine techniques, to improve tissue elasticity and support urethral function.

This study aligns with prior research on the importance of urethral support and elasticity in SUI. For example, dynamic MRI and ultrasonography studies have demonstrated increased bladder neck descent (BND) and urethral hypermobility in women with incontinence. The observed reduction in urethral elasticity is consistent with histological studies showing a decrease in collagen and elastin content in SUI patients. However, the use of SWE in this study represents a methodological advancement by providing quantitative, real-time assessments of urethral stiffness—capabilities that were not previously possible with conventional imaging techniques. These results provide a strong argument for the wider adoption of SWE in both clinical and research environments.

### 4.4. Limitations and Future Directions

While this study provides valuable insights, there are several limitations that need to be considered. The relatively small sample size (n = 30) restricts the ability to generalize the findings, and larger, multicenter studies are needed to validate and expand upon these results. Additionally, the observational nature of the study, which employed a cross-sectional design, precludes drawing causal conclusions. Longitudinal studies will be essential to establish the temporal relationship between changes in urethral elasticity and the progression of SUI. Lastly, the lack of histological correlation between SWE findings and tissue analysis limits the understanding of the molecular mechanisms driving changes in elasticity. Further studies that correlate SWE data with histological examination of urethral tissue could provide more in-depth insights into the underlying pathophysiology.

Future research should explore the role of SWE in monitoring treatment efficacy, such as post-surgical improvements in urethral stiffness. Additionally, studies investigating the interplay between hormonal status, ECM composition, and urethral elasticity could inform novel therapeutic approaches.

## 5. Conclusions

This study highlights significant differences in bladder neck mobility and urethral elasticity between women with SUI and continent controls. The findings underscore the multifactorial nature of SUI and the critical role of urethral support and connective tissue integrity in maintaining continence. By demonstrating the utility of SWE in assessing these parameters, this study paves the way for more personalized, evidence-based approaches to diagnosing and managing SUI.

## Figures and Tables

**Figure 1 bioengineering-12-00373-f001:**
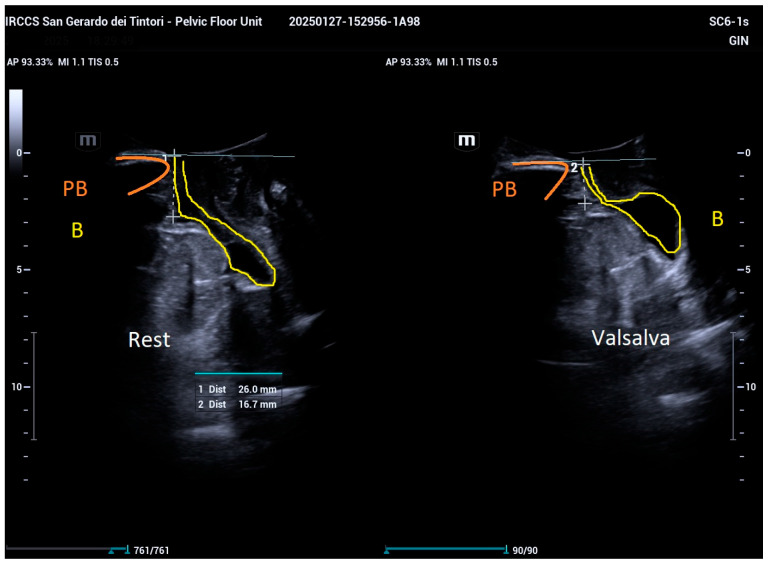
Measurement of bladder neck descent (BND) as the distance between the bladder neck (yellow) and the horizontal axis through the distal symphysis pubis (orange), recorded during rest and maximal Valsalva maneuver.

**Figure 2 bioengineering-12-00373-f002:**
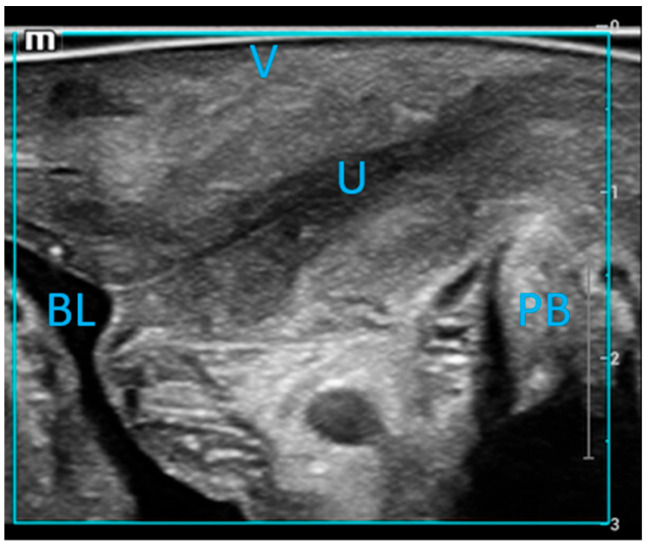
Mid-sagittal imaging of the urethral wall was performed using a linear array probe. The labeled structures include the urethra (U), vagina (V), pubic symphysis (SP), bladder (BL), and pubic bone (PB).

**Figure 3 bioengineering-12-00373-f003:**
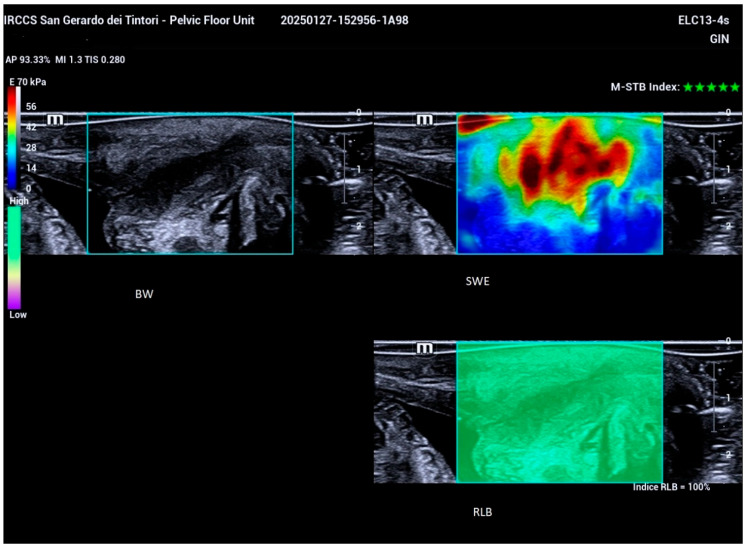
Triple-image real-time display illustrating simultaneous visualization of the 2D ultrasound image (upper left), the corresponding elastography image (upper right), and the RLB Map (lower right), which indicates the reliability of the elastography assessment.

**Figure 4 bioengineering-12-00373-f004:**
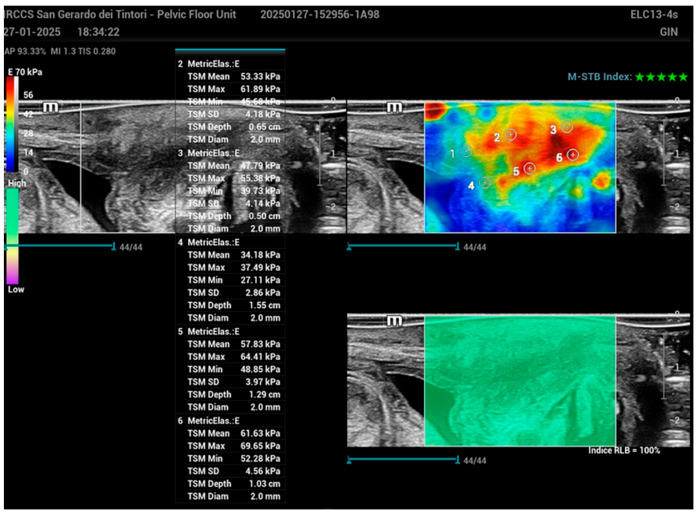
Schematic representation of the urethral wall with regions of interest (ROIs) marked at the proximal, middle, and distal segments in the SWE image. Measurements were taken from six defined regions: the anterior wall approximately 1 cm proximal to the bladder neck, 1 cm from the mid-urethral segment, and 1 cm distal near the external urethral meatus, along with corresponding measurements from the posterior wall at these same locations. ROIs had a diameter of 2 mm for each measurement.

**Table 1 bioengineering-12-00373-t001:** Comparison of BND and Young’s modulus among the groups. Continuous data as mean ± standard deviation. BND: bladder neck descent measured in mm, PA: proximal anterior urethral wall, MA: middle anterior urethral wall, DA: distal anterior urethral wall, PP: proximal posterior urethral wall, MP: middle posterior urethral wall, DP: distal posterior urethral wall, measured in Kpa.

	Group 1	Group 2	*p*-Value
BND (mm)	21.8 ± 7.8	10.5 ± 5	<0.0001
PA (KPa)	29.1 ± 6.4	32.10 ± 6.1	0.12
MA (KPa)	30.7 ± 6.4	57.8 ± 15.6	<0.0001
DA (KPa)	29.9 ± 6.8	47 ± 7.8	<0.0001
PP (KPa)	27.8 ± 5	34.41 ± 6.8	0.0079
MP (KPa)	30.9 ± 5.5	55.9 ± 15.6	<0.0001
DP (KPa)	29.9 ± 7.1	48 ± 10.9	<0.0001

**Table 2 bioengineering-12-00373-t002:** Comparison of Young’s modulus among the groups. P: proximal urethra, M: middle urethra, D: distal urethra.

	Group 1	Group 2	*p*-Value
P (KPa)	28.5 ± 5.7	33.2 ± 6.5	0.0046
M (KPa)	30.8 ± 5.8	56.8 ± 15.4	<0.0001
D (KPa)	29.9 ± 6.8	47.5 ± 9.4	<0.0001

## Data Availability

The data presented in this study are available on request from the corresponding author due to privacy.

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
