# Peer review of "Assessment of Urethral Elasticity by Shear Wave Elastography: A Novel Parameter Bridging a Gap Between Hypermobility and ISD in Female Stress Urinary Incontinence"

_bioengineering, 2025, doi:10.3390/bioengineering12040373_

Round 1
Reviewer 1 Report
Comments and Suggestions for Authors
In Particular:
At least a sample of the original tracing would illustrate the quality and topography of what is a "most important parameter"
The notation used is not in accord to the accepted protocol ie Maximum urethral pressure and function urethral length.
Authors division of the urethral length into what appears arbitrary regions is not accurate in that it appears that the sphincteric aspect of the urethra may be overlapping the arbitrary regions chosen.
The anatomical position of urethral pressure relative to the biomechanics measured would be on inestimable value incorporating the new data with the clinically important aspects of urinary continence.
Author Response
I would like to express my sincere gratitude for the time and effort you dedicated to reviewing this article.
-
This observation is indeed highly relevant, and as you suggested, it provides an excellent opportunity to further enhance the quality of the analysis. The reason this was not implemented is that it requires an invasive procedure, which is not typically performed in an outpatient setting unless the patient presents with mixed incontinence, as is not the case here. Furthermore, such an approach would require the control group to undergo an invasive examination as well, which has not been approved by the ethics committee.
-
The notation employed deviates from the standard accepted by the scientific community, as the parameters in question, such as Young's Modulus, are not directly comparable. In addition to pressure, we reference the parameter of elasticity/rigidity under tension or compression of the urethral tissue.
-
The regions analyzed follow the anatomical division of the distal, middle, and proximal urethra, with the sphincter component specifically evaluated in the middle portion. Thanks to the high-resolution ultrasound equipment used, we are able to distinguish the rhabdomyosphincter in these areas.
-
The integration of urethral pressure measurements with the anatomical position of the urethra and associated biomechanics could significantly enhance the clinical relevance of the data. However, as previously mentioned, this would require an invasive procedure. The primary aim of this study is to explore how ultrasound might serve as a non-invasive alternative, thus avoiding the need for such a procedure while focusing on assessing the elastic component or pressure resistance of the urethra.
Once again, I deeply appreciate your valuable input.
Dr. De Vicari D.
Reviewer 2 Report
Comments and Suggestions for Authors
The manuscript is wide, well written, interesting to the readers. The topic is of interest and is not overtreated. Not many papers treat about the use of ultrasound for incontinence, or urethral strictures and evaluation of urethra, especially for female patients.
Abstract, introducion are clear and well written.
Methods must be implemented. Exclusion criteria are well highlighted. However, inclusion criteria are not well highlighted too.
Line 588-590 : 30 and 15 are already results. Please move them to the result session.
Please rate the experiens in such disease and ultrasould of the examinator
Limitations are well declared, results clearly presented and discussion nicely written.
I would insert some inputs also about what there is on ultrasound use for urethal evaluation in men (i.e. for example for incontinence and stricture) and a comment if it can be reproducible in women
Author Response
Thank you very much for your time and effort spent on this revision work. Below are the corrections made in response to your suggestions:
- A section on inclusion criteria has been added to the Methods section.
- The number of selected patients has been included only in the Results section, as requested.
- The experience of the examiner has been incorporated. Line 345-347
- Regarding the request for input on urethral ultrasound evaluation in the male population, we regret to inform you that this is difficult to address. Our team exclusively handles gynecology, while male evaluations are carried out by a dedicated team of urologists.
Thank you again for your valuable feedback.
De Vicari D.
Reviewer 3 Report
Comments and Suggestions for Authors
A very interesting paper, that has the aim of showing the usefulness of evaluating bladder neck descent (BND) and urethral stiffness through transperineal and intravaginal ultrasonography, incorporating shear wave elastography (SWE) in patients with stress urinary incontinence.
Abstract is very long and appears as a wall of text, making it very difficult to follow, and as such it might deter readers. I ask the authors to refer to the Instructions for Authors of the journal and to shorten the abstract to 200 words maximum.
Introduction chapter is very detailed and offers all the information about stress urinary incontinence. Even though the authors’ intentions were good, it resulted in a 12 pages long detailed description of SUI. Introduction chapter should be brief, 12 pages of text will be too exhaustive for readers. Please review the introduction chapter and shorten is order to make it more accessible.
Material and methods chapter is clear, concise, shows all the methodology involved in evaluating the 30 patients included in the study, and is e everything is throughly explained with pictures, making everything much easier to understand.
Results chapter is clear and everything is presented in a very accessible fashion.
Discussions chapter is comprehensive, the authors present other papers related to the subject. Everything presented in the discussions chapter is relevant to the theme of the paper.
Conclusions chapter is clear and concise.
I appreciate the authors for also presenting the weak points of their paper and the need for further research.
In conclusion even though the paper is very interesting it needs some polishing before being published, so I recommend for rejection for major revision of manuscript text.
Comments on the Quality of English LanguageEnglish language is fine and easy to understand.
Author Response
Thank you very much for your time and effort in reviewing this article. Every aspect has been thoroughly examined, which has greatly facilitated the polishing process.
1-2 The abstract has been revised to meet the 200-word limit as requested, and the introductory section has been shortened as recommended.
We sincerely appreciate your valuable assistance.
De Vicari D.
Round 2
Reviewer 1 Report
Comments and Suggestions for Authors
Authors have made extensive corrections improving the paper.
Clearly some sugggestions could not be adhered to but this would require further unvasive studies not possible under exisiting protocol.
Nonethe less the focus of the study, which is bioengineering, is the ultrasound componnt of the investigation which was both original and most informative.
Author Response
I sincerely appreciate the valuable suggestions provided and the support in reviewing the article. Your insightful feedback and recommendations will undoubtedly be beneficial for future research and further studies. Thank you very much for your time and expertise.
D. De Vicari